# Data Stewardship and Curation Practices in AI-Driven Genomics and Automated Microscopy Image Analysis for High-Throughput Screening Studies: Promoting Robust and Ethical AI Applications

## Abstract

The increasing adoption of AI and next-generation sequencing (NGS) has revolutionized genomics and high-throughput screening (HTS), transforming how cellular processes and disease mechanisms are understood. However, these advancements generate vast datasets requiring effective data stewardship and curation practices to maintain data integrity, privacy, and accessibility. This review consolidates existing knowledge on key aspects, including data governance, quality management, privacy measures, ownership, access control, accountability, traceability, curation frameworks, and storage systems. Major challenges such as managing biases, ensuring data quality, and securing privacy are highlighted. Advanced cryptographic techniques, federated learning, and blockchain technology are proposed as strategic solutions, emphasizing standards compliance, ethical oversight, and tailored access control frameworks. Effective data stewardship is vital for advancing AI-driven genomics and microscopy research. Stakeholders must prioritize robust data governance and privacy measures to ensure data integrity and ethical use. Collaborative efforts should focus on developing transparent data-sharing policies and interoperable platforms to foster innovation and advance research practices. The study promotes collaboration among researchers, robust data governance, privacy and security, clear policies, and educational initiatives to prepare future researchers.

## 1 Introduction

Advancements in AI and next-generation sequencing (NGS) have revolutionized genomics and high-throughput screening (HTS) studies, enabling the integration of multi-dimensional data [6, 4, 5]. Automated high-content screening (HCS) methodologies, combining microscopy image acquisition and analysis, are now pivotal for understanding cellular processes and assessing drug efficacy [3, 2]. However, these technologies generate vast datasets that require robust data stewardship and curation practices to ensure their reliability and accessibility. Therefore, this study aimed to elucidate best practices in data stewardship and curation for AI-driven genomics and automated microscopy image analysis within high-throughput screening studies.

## 2 Methods

A systematic literature search was conducted up to December 30, 2023, across PubMed, MEDLINE, Embase, Scopus, and Web of Science. The search focused on data governance, curation frameworks,

algorithmic bias, and data storage. Realist synthesis methodology was used to integrate diverse theoretical frameworks, with three independent reviewers. The review process included six stages, starting with an extensive search across multiple research databases, resulting in 273 documents. This was followed by screening based on broad criteria, titles, abstracts, and full texts, which narrowed the pool to 38 highly relevant citations.

# 3   Experimental Results

Our findings revealed a significant surge in research activity in recent years, particularly in 2023, reflecting the increasing recognition of the importance of robust data governance frameworks. Notably, while 36 articles extensively discussed data interoperability and sharing measures, areas such as model explainability and data augmentation remained underexplored, highlighting crucial gaps that need to be addressed. The integration of diverse data types, including sequencing, clinical, proteomic, and imaging data, underscored the complexity and breadth of AI applications in genomics and microscopy.

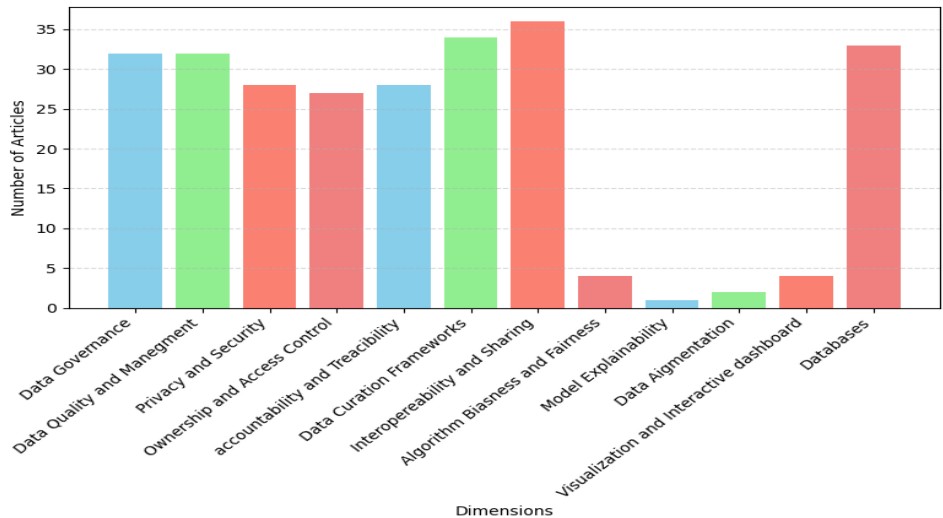

Figure 1: Illustrated the number of articles addressed the data stewardship and curation practice through different dimensions

Moreover, our review emphasized that infrastructure optimization, ethical considerations, access control mechanisms, and transparent data sharing policies were the most critical challenges in AI-based data stewardship. Advanced cryptographic techniques, federated learning, and blockchain technology were proposed to address challenges like data quality, privacy, and bias management. We identified robust data governance measures, such as GA4GH standards[3], DUO versioning, and attribute-based access control[2], as essential for ensuring data integrity, security, and ethical use. The importance of Data Management Plans (DMPs) [1], meticulous metadata curation, and advanced cryptographic techniques emerged as pivotal in mitigating data security and identifiability risks.

# 4   Conclusion

These findings provide a comprehensive overview of current practices and challenges in data stewardship, offering a roadmap for enhancing the robustness and ethical standards of AI applications in genomics and microscopy. Effective data stewardship and curation are vital for advancing AI-driven genomics and microscopy image analysis. Prioritizing robust governance, quality management, and secure sharing frameworks is essential. Collaborative efforts must focus on developing transparent data-sharing policies and interoperable platforms to foster innovation and advance research practices.

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
