# OpenReview forum: "Data Stewardship and Curation Practices in AI-Driven  Genomics and Automated Microscopy Image Analysis  for High-Throughput Screening Studies: Promoting  Robust and Ethical AI Applications"
_IEEE.org/ICIST/2024/Conference — IEEE ICIST 2024 Conference Submission_

### Official Review · Reviewer_ZwQ6 · 2024-08-22
**This article is acceptable after detailed revisions**

**Rating:** 6
**Confidence:** 3

**Review:**

This review comprehensively integrates existing knowledge across key domains such as data governance, quality management, privacy safeguards, ownership rights, access control mechanisms, accountability measures, traceability capabilities, management frameworks, and storage systems. The obtained result is valuable and can be accepted if the following problems can be clarified.
(1)	In the introduction, the shortages of those relevant studies are suggested to be further summarized.
(2)	There exist several spelling and grammar errors. Please check carefully and further polish
(3)	In the Methods and Experimental results, more analysis can be added to better explain the main contents of this paper.
(4)	The future work is missing in the Conclusion.
(5)	The references should be updated and their format standardized for enhanced consistency and accuracy.

---

### Official Review · Reviewer_r3kH · 2024-08-23
**accept**

**Rating:** 7
**Confidence:** 3

**Review:**

The paper comprehensively integrates existing knowledge across key domains such as data governance, quality management, privacy safeguards, ownership rights, access control mechanisms, accountability measures, traceability capabilities, management frameworks, and storage systems. The theory is correct and can be accepted after responding the following comments.
(1)There are many typos and grammar errors. The authors should have a native English speaker or software packages to perform the editing check.
(2)The conclusion suggests adding a section on the prospects for future research.
(3)Please check if you need to update your Introduction.

---

### Official Review · Reviewer_ofx7 · 2024-08-25
**Accept**

**Rating:** 7
**Confidence:** 3

**Review:**

Comment: This review consolidates existing knowledge on key aspects, including data governance, quality management, privacy measures, ownership, access control, accountability, traceability, curation frameworks, and storage systems. The study promotes collaboration among researchers, robust data governance, privacy and security, clear policies, and educational initiatives to prepare future researchers. The theory is correct and can be accepted after responding the following comments.
(1) In the introduction, it is not enough to state the current work. It should be expanded and reconstructed.
(2) There are many typos and grammar errors. The authors should have a native English speaker or software packages to perform the editing check.
(3) Insufficient discussion in the experimental section.

---

### Comment · Reviewer_ZwQ6 · 2024-08-21
**This article is acceptable after detailed revisions**

This review comprehensively integrates existing knowledge across key domains such as data governance, quality management, privacy safeguards, ownership rights, access control mechanisms, accountability measures, traceability capabilities, management frameworks, and storage systems. The obtained result is valuable and can be accepted if the following problems can be clarified.
(1)	In the introduction, the shortages of those relevant studies are suggested to be further summarized.
(2)	There exist several spelling and grammar errors. Please check carefully and further polish
(3)	In the Methods and Experimental results, more analysis can be added to better explain the main contents of this paper.
(4)	The future work is missing in the Conclusion.
(5)	The references should be updated and their format standardized for enhanced consistency and accuracy.

---

### Decision · Program_Chairs · 2024-09-06

Accept (Oral)